# Cytokine Storms and Anaphylaxis Following COVID-19 mRNA-LNP Vaccination: Mechanisms and Therapeutic Approaches

**DOI:** 10.3390/diseases12100231

**Published:** 2024-10-01

**Authors:** Toru Awaya, Hidehiko Hara, Masao Moroi

**Affiliations:** 1Department of Cardiovascular Medicine, Toho University Ohashi Medical Center, 2-22-36, Ohashi Meguro-ku, Tokyo 153-8515, Japan; 2Department of Internal Medicine, Misato Central General Hospital, Saitama 341-8526, Japan

**Keywords:** mRNA vaccine, adverse reaction, inflammatory cytokine, lipid nanoparticles, corticosteroid, interleukin 6, cytokine storm, anaphylaxis, Kounis syndrome

## Abstract

Acute adverse reactions to COVID-19 mRNA vaccines are a major concern, as autopsy reports indicate that deaths most commonly occur on the same day of or one day following vaccination. These acute reactions may be due to cytokine storms triggered by lipid nanoparticles (LNPs) and anaphylaxis induced by polyethene glycol (PEG), both of which are vital constituents of the mRNA-LNP vaccines. Kounis syndrome, in which anaphylaxis triggers acute coronary syndrome (ACS), may also be responsible for these cardiovascular events. Furthermore, COVID-19 mRNA-LNP vaccines encompass adjuvants, such as LNPs, which trigger inflammatory cytokines, including interleukin (IL)-1β and IL-6. These vaccines also produce spike proteins which facilitate the release of inflammatory cytokines. Apart from this, histamine released from mast cells during allergic reactions plays a critical role in IL-6 secretion, which intensifies inflammatory responses. In light of these events, early reduction of IL-1β and IL-6 is imperative for managing post-vaccine cytokine storms, ACS, and myocarditis. Corticosteroids can restrict inflammatory cytokines and mitigate allergic responses, while colchicine, known for its IL-1β-reducing capabilities, could also prove effective. The anti-IL-6 antibody tocilizumab also displays promising treatment of cytokine release syndrome. Aside from its significance for treating anaphylaxis, epinephrine can induce coronary artery spasms and myocardial ischemia in Kounis syndrome, making accurate diagnosis essential. The upcoming self-amplifying COVID-19 mRNA-LNP vaccines also contain LNPs. Given that these vaccines can cause a cytokine storm and allergic reactions post vaccination, it is crucial to consider corticosteroids and measure IL-6 levels for effective management.

## 1. Introduction

Autopsy data have shown that deaths most frequently occur on the same or next day following the administration of COVID-19 mRNA-lipid nanoparticle (LNP) vaccines, with cardiovascular events accounting for approximately half of the deaths [1]. While these adverse events are typically attributed to allergic reactions with conventional vaccines, COVID-19 mRNA-LNP vaccines have been associated with an incidence of deaths due to cytokine storms [2,3,4,5]. These events are thought to be driven by cytokine storms induced by LNPs and the spike proteins themselves [6,7,8] or allergic responses such as Kounis syndrome [9] (Figure 1). Additionally, the SARS-CoV-2 spike protein binds to angiotensin-converting enzyme 2 (ACE2) and downregulates its expression, which increases the level of angiotensin II (AngII) [10]. AngII and IL-1β upregulate the activity of Rho-kinase, resulting in hypersensitivity to vascular smooth muscle contraction [11] (Figure 1).

Recent reports have indicated that frequent mRNA vaccination may induce a gradual increase in TRAb levels and promote a class switch to IgG4, which could contribute to long-term immune responses and might be considered potential late-onset adverse reactions [12,13]. Additionally, it has been suggested that the use of 100% N1-methyl-pseudouridine (m1Ψ) in mRNA vaccines may pose a potential risk of promoting cancer development in the long term, as m1Ψ has been associated with immune suppression and the facilitation of tumor growth and metastasis in certain models [14].

In this review, we summarize the cytokine release and allergic reactions, including Kounis syndrome, associated with COVID-19 mRNA-LNP vaccines and recommend several precautionary measures and treatments, including corticosteroids and colchicine.

## 2. Mechanism, Treatment, and Prevention of Cytokine Release Following COVID-19 mRNA-LNP Vaccination

### 2.1. Mechanism of Cytokine Release Following COVID-19 mRNA-LNP Vaccination

In COVID-19 mRNA-LNP vaccines, ionizable lipids are used in LNPs instead of cationic lipids. Negatively charged mRNA requires positively charged cationic lipids for uptake by cells and escape from the endosomes. However, cationic lipids stimulate the excessive production of inflammatory cytokines [15]. To address this, ionizable lipids which become positively charged only in acidic environments (such as within the endosomes) were developed. These ionizable lipids have reduced potential to stimulate the production of inflammatory cytokines compared with cationic lipids [16]. However, during the process of endosomal escape, ionizable lipids disrupt the endosomal membrane to release mRNA, which can, in some cases, lead to the excessive production of inflammatory cytokines [17]. Indeed, gene analyses have shown that ionizable lipids activate various inflammatory pathways, including those involved in viral infections, RIG-I, and Toll-like receptor signaling, resulting in increased levels of IL-1β and IL-6. LNPs without ionizable lipids do not elicit any visible signs of inflammation [6] (Figure 1 and Figure 2). Furthermore, the cationic lipid ALC-0315, used in mRNA vaccines, has an apparent pKa of 6.09, which is lower than the optimal range for intramuscular administration (6.6–6.9). This suboptimal pKa contributes to instability in mRNA delivery, potentially reducing vaccine efficacy and increasing the risk of inflammatory responses [17].

The immunostimulatory effects of LNPs depend on the presence of ionizable lipids, with IL-6 induction playing a critical role in their adjuvant activity [18], while mRNA itself may also act as an adjuvant, potentially contributing to the risk of cytokine storms. This occurs because when melanoma differentiation-associated gene 5 (MDA5) recognizes mRNA, it enhances interferon responses and activates CD8+ T cells. Studies have shown that in MDA5-deficient mice, interferon responses induced by mRNA-LNP vaccines are significantly impaired [19]. Although the m1Ψ modification prevents mRNA from being sensed by endosomal TLR7, vaccine-derived mRNA may still contribute to innate immune responses and severe inflammation [20].

Autopsy cases following vaccination showed cytokine storms in patients whose genetic tests revealed upregulated cytokine signaling [2]. Ionizable lipids in LNPs are also components of small interfering RNA (siRNA) therapeutics, such as patisiran. Elevated inflammatory cytokine levels have also been reported after administration of patisiran. Therefore, premedication with a corticosteroid such as dexamethasone or histamine (H)1 and H2 blockers is required before infusion [21]. Cytokine release due to LNPs can also manifest as nausea and coughing, making it challenging to distinguish these from allergic reactions [22]. A cytokine storm caused by LNPs can be mistakenly identified as anaphylaxis, in which case the patient may be administered epinephrine. However, epinephrine may also increase IL-6 levels and potentially worsen outcomes during a cytokine storm [23]. Therefore, when it is difficult to determine whether a patient is experiencing a cytokine storm or anaphylaxis, the patient must be administered corticosteroids in addition to epinephrine (Figure 2).

### 2.2. Treatment of Cytokine Release Following COVID-19 mRNA-LNP Vaccination

Management guidelines for each grade of cytokine release syndrome following chimeric antigen receptor T-cell (CAR-T) therapy and immune checkpoint inhibitors have been published [24,25]. Grade 2 toxicity resulting from CAR-T therapy is characterized by fever (≥38.0 °C), hypotension not requiring vasopressors, hypoxia requiring low-flow oxygen via nasal cannula (up to 6 L/min), and organ dysfunction. In contrast, grade 3 toxicity exhibits more severe symptoms which require vasopressors to regulate hypotension and oxygen supplementation exceeding 6 L/min. Grade 4 represents the most severe toxicity level, which requires the use of positive pressure ventilation, such as mechanical ventilation, to support a patient’s respiration. The anti-IL-6 antibody tocilizumab is reported to be effective for managing such toxicity of grade 2 or higher [24,26]. The dose of corticosteroids should also be based on the severity of the condition (e.g., 40 mg of dexamethasone for grade 2, 40–80 mg for grade 3, and steroid pulse therapy for grade 4). Therefore, steroid therapy may also be combined with tocilizumab for managing cytokine storms following vaccination [27,28] (Figure 2).

Corticosteroids, such as dexamethasone, are highly effective, owing to their capability in suppressing inflammation at the molecular level. When glucocorticoids, a specific subclass of corticosteroids, enter the cell, they bind to and activate the glucocorticoid receptor (GR). Subsequently, the activated GR translocates into the nucleus and binds to specific DNA regions, inhibiting the transcription of pro-inflammatory genes [29,30]. Moreover, corticosteroids reduce Toll-like receptor signaling by blocking key transcription factors such as NF-κB and AP-1, thereby blunting the expression of various pro-inflammatory cytokines, including those in the IL-1 to IL-6 family [31]. The action of corticosteroids on macrophages and mast cells blocks the production of histamine and eicosanoid-lipid mediators such as prostaglandin E2 and leukotriene B4 [31]. This vital mechanism renders corticosteroids highly effective in reducing inflammatory responses and allergic reactions [29,30,31] (Figure 2). The anti-IL-6 antibody tocilizumab affects the inflammatory response and diminishes the production of plasminogen activator inhibitor-1 (PAI-1), which is linked to coagulation and thrombosis [28,32]. The reduced PAI-1 levels can lower the risk of thrombosis, which is particularly relevant under elevated IL-6 conditions, such as severe COVID-19 and other forms of cytokine release syndrome. However, it is essential to note that tocilizumab hinders the binding of IL-6 to its receptors. Hence, an increase in serum IL-6 levels is expected when measuring IL-6 levels after the administration of tocilizumab. Given that a heightened IL-6 level reflects the drug’s mechanism instead of disease progression, caution is necessary when interpreting these elevated levels in this context [32]. Additionally, in a single-case study, tocilizumab has been shown to effectively treat new-onset rheumatoid arthritis by reducing elevated serum IL-6 levels following COVID-19 mRNA-LNP vaccination [33].

There has been growing attention over recent developments in the use of colchicine, given its superior effectiveness for treating gout and autoinflammatory diseases compared with traditional applications [34]. Notably, colchicine has been shown to suppress formation of the NOD-like receptor family pyrin domain-containing 3 (NLRP3) inflammasome [35], a crucial mediator in the inflammatory response stimulated by ionizable LNPs and spike proteins [6,8], which may subsequently reduce IL-1β levels. This mechanism indicates the potential use of colchicine as a valuable therapeutic option in specific scenarios requiring inflammation control. Aside from this, recent studies have demonstrated that the use of colchicine instead of aspirin in ACS does not exacerbate thrombotic events, signifying its favorable effects on platelet function and inflammatory profiles [36]. Moreover, colchicine has been proven to minimize severe cardiovascular events and cardiovascular fatality in patients with stable coronary disease [37]. Considering these findings, the United States Food and Drug Administration in 2023 approved colchicine for clinical use at a daily dosage of 0.5 mg in patients with established atherosclerotic disease or multiple cardiovascular risk factors [34].

Evidence suggesting the effectiveness of colchicine monotherapy in treating myocarditis cases linked to the administration of bivalent COVID-19 mRNA-LNP vaccines has also emerged, particularly those involving high levels of cytokine release. However, this promising benefit requires further research and confirmation [38] (Figure 2). Since IL-1β is involved in immune activation and antibody production [39], the use of colchicine may potentially suppress these functions, leading to a slightly higher risk of infections, such as pneumonia [40]. For instance, in a 22 month follow-up, the prevalence of pneumonia was more than two times higher in the colchicine group compared with the control group (0.9% versus 0.4%, respectively) [40]. Despite posing a minor risk, it is imperative to exercise caution when administering colchicine to patients with compromised immune systems or those at higher risk of infection. Therefore, it is crucial to consider an appropriate follow-up prior to using colchicine, including imaging studies, such as chest X-rays, to ensure early detection and proper management of potential infections.

### 2.3. Prevention of Cytokine Release Following COVID-19 mRNA-LNP Vaccination

Bathing has been recognized to promote the release of cytokines, such as IL-6 [41], as recent reports have highlighted deaths occurring shortly after vaccination during bathing in a bathtub [42]. Alternatively, showering may be a safer cleansing option immediately after vaccination. Apart from that, high-intensity exercise can significantly boost expression of the NLRP3 gene and IL-1β. By contrast, chronic moderate-intensity exercise suppresses these markers [43]. Given that high-intensity exercise may promote higher LNP-derived IL-1β levels, it is advisable to avoid such exercise regimes post vaccination. Alcohol consumption also increased IL-6 levels three hours after initial intake. Hence, excessive drinking after vaccination should also be avoided [44].

## 3. IL-6 Levels and Symptoms Underlying Cytokine Release Following COVID-19 mRNA-LNP Vaccination

LNPs trigger the production and release of inflammatory cytokines, particularly IL-6 and IL-1β, due to their ionizable lipid component [6] (Figure 1 and Figure 2). A case of the death of a 14 year-old girl was published, in which the cause of death was reported to be multi-organ inflammation two days after receiving the COVID-19 mRNA-LNP vaccine [3]. In this case, IL-6 levels were markedly elevated to 226 pg/mL (normal ≤ 4 pg/mL), but C-reactive protein (CRP) levels were only slightly elevated to 0.910 mg/dL (normal ≤ 0.3 mg/dL). In another case, an 81 year-old man developed myocarditis after receiving the bivalent COVID-19 mRNA-LNP vaccine. He was also reported to experience hypercytokinemia, with an elevated IL-6 level of up to 95.7 pg/mL. Despite his advanced age, colchicine treatment without steroids significantly improved his IL-6 levels to 3.5 pg/mL. The use of IL-6 may also be beneficial in monitoring treatment response [38]. Serum IL-6 levels have been strongly correlated with cytokine storms and myocarditis. Studies have also revealed a strong correlation between IL-6 levels and specific conditions, such as sepsis, acute respiratory distress syndrome, and COVID-19 infection [32]. Therefore, it may be advantageous to monitor IL-6 levels in cases of unexplained fever. The IL-6 level is also linked to CRP, although the IL-6 levels’ increase substantially during a cytokine storm [24,45]. It is not clear why some patients develop a severe cytokine storm, but genetic variations in IL-6 may be a contributing factor [25]. The evidence thus points to a greater role played by IL-6 than CRP in the development of acute cytokine storms.

Symptoms which might suggest excessive cytokine release include a body temperature ≥ 38 °C, hypotension with systolic blood pressure < 90 mmHg, hypoxia with SpO_2_ levels ≤ 90% on room air, or evidence of organ toxicity [24]. Organ toxicity includes cardiac symptoms such as tachycardia, arrhythmia, and a low ejection fraction. Respiratory symptoms include tachypnoea and pulmonary edema. Hepatic symptoms involve increased serum aspartate aminotransferase and alanine aminotransferase levels. Renal symptoms include elevated creatinine and decreased urine output. Dermatological symptoms such as rashes can occur.

Cardiac symptoms are prevalent side effects following COVID-19 mRNA-LNP vaccination, rendering their evaluation crucial [46]. This is due to the strong association between myocarditis and ACS with inflammatory cytokines, such as IL-1β and IL-6 [47,48]. Therefore, the use of multimodality approaches, including electrocardiogram (ECG), echocardiography, cardiac magnetic resonance imaging, and high-sensitivity troponin T [49,50], is essential to ensure these conditions are not overlooked [42]. Thus, it is necessary to monitor for potential infections, particularly if symptoms such as fever develop. Furthermore, it was suggested that cytokine responses to various bacterial pathogens, such as *Escherichia coli*, *Haemophilus influenzae*, and *Staphylococcus aureus*, as well as non-COVID-19 viruses decrease one month after vaccination [51], which indicates a potential temporary reduction in immune function post vaccination. Therefore, it is vital to evaluate blood cultures, conduct imaging studies, and perform procalcitonin tests, as well as measure IL-6 levels to rule out possible infections. Additionally, when both cytokine storms and sepsis are very likely to occur, the initiation of empirical antibiotic and steroid therapy should be considered due to the high risk of rapid deterioration. Furthermore, vitamin C (ascorbic acid), due to its antioxidant, anti-inflammatory, and immunomodulatory properties, has been suggested to be potentially beneficial in the treatment of cytokine storms associated with COVID-19. Specifically, high-dose intravenous vitamin C administration may reduce oxidative stress and lower levels of pro-inflammatory cytokines such as IL-6, thereby possibly mitigating the severity of cytokine storms. However, the current evidence supporting this is limited, and further research is necessary [52].

## 4. Mechanisms of Allergic Reactions to COVID-19 mRNA-LNP Vaccines

Among the 1,893,360 individuals who received their first dose of the Pfizer-BioNTech COVID-19 vaccine, 175 cases of severe allergic reactions were reported to the Vaccine Adverse Event Reporting System. Of these, 21 cases were classified as anaphylaxis. Among those who experienced anaphylaxis, 17 had a documented history of allergies or allergic reactions, and 7 had a previous history of anaphylaxis [53]. COVID-19 mRNA-LNP vaccines assemble into LNPs using additional neutral lipids, cholesterol, and a phospholipid conjugated to PEG. Allergic reactions to neutral lipids, cholesterol, PEG, and PEG-conjugated phospholipids contained in LNPs have been reported [54,55]. Therefore, patients with a history of severe allergic reactions to PEG or polysorbate, which has a similar structure to PEG, should avoid COVID-19 mRNA-LNP vaccination.

While these reactions may involve immunoglobulin E (IgE)-mediated mechanisms, non-IgE pathways, such as complement activation-related pseudoallergy (CARPA), are also implicated. CARPA is characterized by complement system activation, which leads to mast cell degranulation without the involvement of IgE. This process results in symptoms resembling traditional allergies, including vasodilation, increased vascular permeability, and immune cell recruitment. In studies investigating CARPA related to COVID-19 mRNA vaccines, a porcine model has been employed, where some animals exhibited tachyphylaxis, a phenomenon in which subsequent doses failed to provoke a reaction following an initial anaphylactic shock. However, it is important to note that this tachyphylactic response has been observed in animal models, and its relevance to human immune responses remains unconfirmed [56].

Furthermore, activation of the Mas-related G protein-coupled receptor X2 (MRGPRX2) can lead to severe hypersensitivity reactions, bypassing the need for specific IgE or elevated tryptase levels. This alternative pathway may explain the hypersensitivity reactions observed in some vaccine recipients. In addition to immediate hypersensitivity reactions, type IV hypersensitivity (delayed-type reactions) can occur 48 h post vaccination, with symptoms peaking between 72 and 96 h. These reactions are mediated by T-cells and monocytes or macrophages, resulting in cytokine release and tissue inflammation. Although less common, these delayed responses should be considered in patients presenting with symptoms several days after vaccination [54].

Recent studies suggest that individuals with significantly elevated levels of anti-PEG antibodies, termed ‘anti-PEG antibody supercarriers’, are at higher risk of hypersensitivity reactions, including anaphylaxis, upon exposure to PEG-containing COVID-19 vaccines such as Comirnaty and Spikevax [57]. Approximately 3–4% of the general population falls into this category, possessing 15–45 times higher levels of anti-PEG antibodies compared with the general population. In recipients of Spikevax, an increase in anti-PEG antibodies has been confirmed in all vaccinated individuals, while in recipients of Comirnaty, approximately 10% exhibited a significant rise in antibody levels. Complement activation following PEG exposure may further exacerbate immune reactions, contributing to CARPA, which has been associated with the breakdown of LNPs and a reduction in vaccine efficacy due to accelerated blood clearance and systemic inflammation. Screening for the presence of anti-PEG antibody supercarriers before vaccination could potentially mitigate the risk of anaphylaxis and other hypersensitivity reactions. In light of the high number of deaths within a few days post vaccination, strict monitoring during this period is crucial.

## 5. Kounis Syndrome Induced by Allergic Reactions to COVID-19 mRNA-LNP Vaccines

### 5.1. Mechanism of Kounis Syndrome

Allergic reactions to COVID-19 mRNA-LNP vaccines can occasionally lead to allergic ACS, also referred to as Kounis syndrome [58] (Figure 1). This syndrome is categorized into four variants. Type I relates to coronary spasms in normal or nearly normal coronary arteries; type II includes ACS with plaque erosion or rupture due to spasms; type III encompasses restenosis and obstruction of stents; and type IV involves thrombosis of coronary artery bypass grafts [59]. The coronary spasm in type I is deemed the most common variant, representing about 70% of cases [60]. Cases of coronary spasms and stent thrombosis have been reported post mRNA-LNP vaccination [9,61]. Hence, particular caution should be given when administering COVID-19 mRNA-LNP vaccines, as reports have shown that they increase the risk of thrombosis [62] (Figure 1), with a heightened incidence of myocardial infarction one day after vaccination [63]. Kounis syndrome can be triggered by a wide range of substances, including various pharmaceutical agents, such as antibiotics [64], contrast media, and vaccines [9,61], as well as non-pharmaceutical compounds, including food allergies, metal allergies, and stings from insects [65].

Kounis syndrome is mainly associated with the synthesis of inflammatory mediators, including histamine, tryptase, chymase, and arachidonic acid, which trigger coronary artery spasms [58,66]. Histamine exerts varying effects on coronary arteries, depending on the physiological state of the endothelium and the histamine concentration. Histamine serves as a vasodilator in normal or nearly normal coronary arteries. Nevertheless, histamine can cause excessive vasoconstriction in patients with impaired endothelia. The activity of histamine through the H1 receptor is also concentration-dependent. In other words, a low histamine dosage stimulates vasodilation, while high doses lead to vasoconstriction [60]. Meanwhile, tryptase is vital for diagnosing anaphylaxis and myocardial ischemia, as the serum tryptase levels typically peak within an hour after symptom onset [59,67]. Therefore, measuring tryptase levels is recommended for patients suspected of experiencing Kounis syndrome [68]. Aside from this, histamine released from mast cells and platelet activation through Fc receptors interacting with mast cells serve vital roles in inflammation. Taking into account the ability of platelet-activating factor (PAF) to induce inflammation and promote platelet aggregation, it is crucial to inhibit mast cell degranulation and suppress PAF [69]. Interestingly, rupatadine is able to block histamine H1 receptors and antagonize PAF receptors. This dual action can potentially mitigate coronary vasoconstriction caused by histamines and impede platelet accumulation and inflammation induced by PAF [70], thereby modulating both major pathways involved in Kounis syndrome.

### 5.2. Diagnosis and Treatment of Kounis Syndrome

The possibility of experiencing Kounis syndrome after vaccination should be considered [9,61]. In cases of chest symptoms, one should immediately perform an ECG examination. Other diagnostic measures, such as blood tests for troponin, creatine kinase (CK), CK-MB, and tryptase, as well as echocardiography should also be conducted. Tryptase is reported to be effective, given its rapid increase within an hour in both anaphylaxis and ACS [59,67]. Patients with ECG trends indicating ACS should be transferred to a facility which can perform emergency catheterization. If their coronary angiography results are normal, then they should consider undergoing an acetylcholine provocation test to exclude the possibility of coronary artery spasms [61]. Additionally, myocardial ischemic events caused by spasms can be evaluated through early beta-methyl-p-iodophenyl-pentadecanoic acid scintigraphy [71].

As a β2-adrenoceptor (β2-AR) agonist, epinephrine is highly effective at inhibiting IgE-dependent mast cell histamine release, offering strong anti-allergic effects [72]. Nevertheless, its alpha (α)-adrenergic effect can trigger coronary artery spasms [73,74]. A total of nine case reports have documented myocardial ischemia following the administration of epinephrine for anaphylaxis [73]. Epinephrine should be administered without delay upon meeting the stipulated criteria for diagnosing anaphylaxis, including life-threatening airway, breathing, circulation, or gastrointestinal problems. In patients with suspected myocardial ischemia, careful treatment options should be considered, including the use of vasodilators such as calcium channel blockers, nitrates, and nicorandil [75]. Notably, studies have reported on precisely adjusting low doses of epinephrine and nicorandil for treating both anaphylaxis and coronary artery lesions [74]. H1 and H2 receptor blockers alleviate allergic symptoms, mitigating complications associated with Kounis syndrome [75]. Additionally, corticosteroids may be part of the treatment protocol to effectively manage both anaphylaxis and potential coronary complications [73,74,75,76]. On the contrary, β-blockers should be avoided in Kounis syndrome, as they can worsen coronary artery spasms due to unopposed α-adrenergic receptor activity [75].

### 5.3. Medical History of Kounis Syndrome

Assessing medical history is also critical to determine the appropriate treatment for Kounis syndrome. Patients with a history of coronary stent placement or bypass surgery are highly susceptible to the type III (stent restenosis and obstruction) and type IV (coronary artery bypass graft thrombosis) variants [59]. Individuals who carry the aldehyde dehydrogenase 2 (ALDH2)*2, which is associated with deficient ALDH2 activity, are also at a higher risk of coronary artery spasms. This is particularly relevant for those who experience hot flashes (facial flushing) after consuming alcohol, as it may indicate a possible ALDH2*2 carrier status [77]. This necessitates careful monitoring and consideration in the context of Kounis syndrome. Thus, these individuals are advised to abstain from alcohol intake and smoking after vaccination.

Furthermore, alcohol is known to enhance the release of endogenous histamine, which reversibly mitigates the activity of gastrointestinal diamine oxidase (DAO) for a brief period. DAO is responsible for breaking down histamine, and impaired DAO activity can lead to histamine build-up in the body, potentially resulting in a condition known as histamine intolerance (HIT) [78,79,80]. Hence, alcohol consumption immediately after vaccination should be avoided to prevent the accumulation of histamine. Other factors which disrupt DAO activity include mineral and vitamin deficiencies (such as vitamin C and copper), intake of specific medications (such as verapamil, clavulanic acid, and isoniazid), the menstrual cycle, and gastrointestinal diseases (such as inflammatory bowel disease and non-celiac gluten sensitivity) [79]. Individuals with HIT are also recommended to avoid histamine-rich foods, such as cheese, red wine, fermented foods, spinach, and canned fish, after vaccination [80]. Moreover, atopy is linked to a doubled risk of anaphylaxis, emphasizing the importance of exercising extra caution in patients with atopy, regardless of the specific medication group involved. In addition, patients with mast cell disease should be given pre-medication with a histamine H1 receptor blocker, such as cetirizine (10 mg for adults), 1 h prior to vaccination, as recommended [54]. Since histamine is also involved in IL-6 secretion [81], the use of histamine H1 receptor blockers can inhibit the histamine-induced production of IL-6 [82] (Figure 1 and Figure 2).

Kounis syndrome is more likely to occur when β2-ARs are blocked but not β1-ARs. Blocking β2-ARs can intensify cardiac anaphylaxis through increased coronary vasoconstriction and diminished cardiac contractility [83]. Therefore, patients should be mindful of the heightened risk of Kounis syndrome following vaccination before taking β-blockers, especially nonselective β-blocker antagonists such as propranolol [61]. Moreover, patients who consume β-blockers may respond poorly to epinephrine administered during anaphylaxis and may require glucagon as an alternative treatment [84] (Figure 2). Meanwhile, corticosteroids offer beneficial impacts for treating refractory vasospastic angina, particularly in patients with allergic tendencies, such as bronchial asthma. These findings suggest that local inflammation in the coronary arterial wall could trigger coronary spasms, potentially resulting in arterial hyperreactivity [76]. However, corticosteroids containing succinate esters (hydrocortisone sodium succinate, methylprednisolone sodium succinate, and prednisolone sodium succinate) have been reported to trigger high-frequency airway hypersensitivity reactions in patients with aspirin-induced asthma (nonsteroidal anti-inflammatory drug-exacerbated respiratory disease).

## 6. Discussion

The COVID-19 mRNA-LNP vaccine was the first vaccine to utilize LNPs, which was developed for mRNA delivery since 2015 [85]. LNPs were first used worldwide as a drug delivery system in siRNA therapeutics to treat hereditary amyloidosis. For instance, the siRNA therapeutic Patisiran has been shown to elevate inflammatory cytokines levels after infusion. Therefore, patients receiving Patisiran typically require pre-medication with corticosteroids and histamine H1 and H2 receptor blockers [21]. Similarly, the adverse impacts of mRNA-LNP vaccines, especially cytokine release associated with LNPs, should be precisely recognized and addressed (Figure 1 and Figure 2). Second-generation siRNA therapeutics incorporate N-acetylgalactosamine as the delivery system instead of LNPs, eliminating the need for pre-medication with corticosteroids [86]. In one study, jet injectors were used for mRNA delivery in mice instead of LNPs [87]. Remarkably, mice receiving COVID-19 vaccines via jet injectors recorded diminished inflammatory cytokine levels in the lymph nodes, spleen, and liver, as well as decreased lung tissue damage in COVID-19 infected mice.

Despite the exceptional performance of these delivery systems, LNPs are expected to remain in use until the practicality of jet injectors has been comprehensively investigated. Intriguingly, a self-amplifying mRNA (sa-mRNA) vaccine was recently approved in Japan in 2024. An sa-mRNA vaccine functions by encoding both the target antigen—typically the SARS-CoV-2 spike protein—and an enzyme called replicase. The mRNA in the vaccine instructs the cells to produce the initial spike protein. Then, replicase, or RNA-dependent RNA polymerase, facilitates a crucial role by enabling the mRNA to replicate multiple times within the transfected cells. This amplification increases spike protein production, allowing a smaller initial dose of mRNA to trigger a robust immune response, reinforcing vaccine efficiency [88]. As the sa-mRNA vaccine uses LNPs, similar precautions are needed due to the risk of cytokine release triggered by LNPs, the spike protein, and mRNA [6,8,20,89] (Figure 1). Since a history of COVID-19 infection can heighten cytokine levels and increase the risk of myocarditis after COVID-19 mRNA-LNP vaccination [7,90], special caution is warranted when administering the vaccine shortly after infection. The Centers for Disease Control and Prevention recommends a three-month interval between COVID-19 infection and vaccination [91].

In light of this, managing cytokine storms and allergic reactions following vaccination is essential, and the use of corticosteroid therapy and tocilizumab is recommended [24]. Additionally, the increasing interest in the anti-inflammatory properties of colchicine may be beneficial in specific clinical scenarios [38]. Nevertheless, in-depth research is necessary to understand the impact of these treatments on the immune response and the risk of infection. Based on these findings, it is crucial to establish effective management strategies to minimize the risk of cytokine storms and anaphylaxis associated with COVID-19 mRNA-LNP vaccines. Future studies are expected to develop more effective and safer therapeutic and preventive measures.

## 7. Conclusions

This study summarized the mechanisms underlying cytokine release and allergic reactions associated with COVID-19 mRNA-LNP vaccines and discussed relevant preventive measures and treatments. The administration of corticosteroids may be considered to distinguish between cytokine storms and anaphylaxis post vaccination, and the monitoring of IL-6 levels may provide a more accurate diagnosis and appropriate treatment of adverse effects from the COVID-19 mRNA-LNP vaccine.

## Figures and Tables

**Figure 1 diseases-12-00231-f001:**
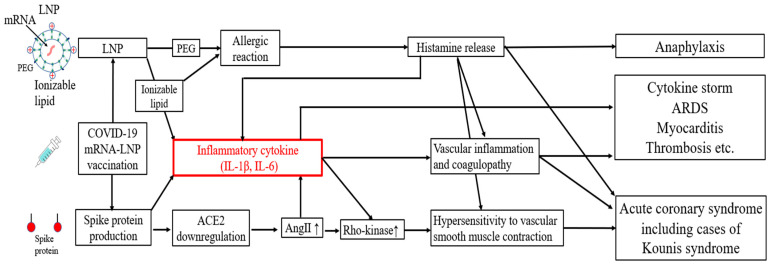
Potential mechanisms of acute adverse effects following COVID-19 mRNA-LNP vaccination. mRNA = messenger RNA; LNP = lipid nanoparticle; PEG = polyethylene glycol; IL = interleukin; ACE2 = angiotensin-converting enzyme 2; Ang II = angiotensin II; ARDS = acute respiratory distress syndrome.

**Figure 2 diseases-12-00231-f002:**
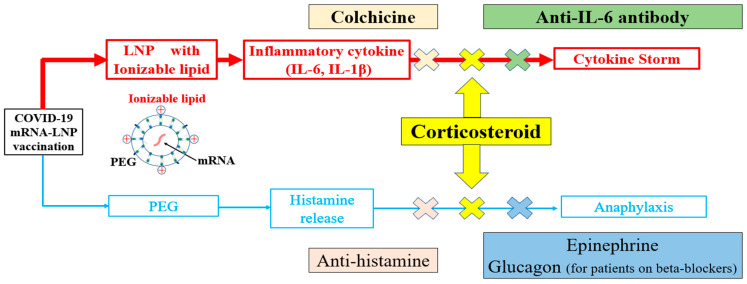
Specific treatments for immediate and short-term reactions to COVID-19 mRNA-LNP vaccinations. Contrary to conventional vaccines, treatment strategies following the administration of COVID-19 mRNA-LNP vaccines must consider allergic reactions and the cytokine release mechanism. In light of the challenges of distinguishing between a cytokine storm and anaphylaxis, corticosteroids are recommended as a treatment option to address both conditions. mRNA = messenger RNA; LNP = lipid nanoparticle; PEG = polyethylene glycol; IL = interleukin.

## Data Availability

Interim Clinical Considerations for Use of COVID-19 Vaccines in the United States under CDC Vaccines & Immunizations (https://www.cdc.gov/vaccines/covid-19/clinical-considerations/interim-considerations-us.html; accessed 30 July 2024).

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
