# Peer review of "Cytokine Storms and Anaphylaxis Following COVID-19 mRNA-LNP Vaccination: Mechanisms and Therapeutic Approaches"

_diseases, 2024, doi:10.3390/diseases12100231_

Round 1
Reviewer 1 Report
Comments and Suggestions for Authors
In this paper, the authors reviewed the literature on adverse event following mRNA-LNP vaccination, in particular focusing on cytokine storms and anaphylaxis. As the authors pointed out, a forensic autopsy case of a 14-year-old girl in Japan who died of multi-organ inflammation two days after mRNA-LNP vaccination has been reported. It is crucial to understand the mechanisms of mRNA-LNP-induced cytokine storms and to investigate preventive and therapeutic options, so this review should be published and read as soon as possible.
Some points that could be improved or addressed are as follows.
#1
Page 2,
The requirement of ionizable lipids for the adjuvant activity of LNP2 has also been shown in basic studies with mice; some articles published in high-impact journals (e.g. Alameh et al, Immunity 54, 2877-2892, 2021 ) should be cited.
#2
Page 2,
What do you think about the possibility that the RNA itself, in addition to the lipids contained in the LNP, may also act as an adjuvant contributing to the risk of cytokine storms? In mice deficient in sensing intracellular dsRNA, the mRNA-LNP vax-induced interferon responses were markedly impaired (Li et al, Nat Immunol 23, 543-555, 2022). So the vaccine-derived mRNA may also contribute to innate immune responses and severe inflammation (even though the N1-methyl-pseudouridine modification prevents mRNA from being sensed by endosomal TLR7) (Kobiyama & Ishii, Nat Immunol 23,472-482,2022). Is it possible that adverse events due to severe inflammation may be more severe for the self-amplifying RNA-LNP vaccine described in Discussion section (page 8)?
Comments on the Quality of English Language
This manuscript is well written. No particular grammatical or spelling errors were found.
Reviewer 2 Report
Comments and Suggestions for Authors
This review covers a current and important topic. The paper is well written and therefore no objection (minor or major) to the presented material. I have another concern though. The paper claims to disclose the mechanisms and therapeutic opportunities in the field. However, I feel deficiencies here. Only a part of the literature, relevant for COVID vaccine-related cytokine storm and anaphylaxis is covered. Most examples are taken from clinical case studies, and possible mechanisms deduced from those. Yet at the end, the authors mention an experimental method for vaccination, and in the Discussion they call for further studies to develop more effective therapeutic and preventive measures. Such studies exist, some research groups in the EU are widely involved in that. However, most of them will not be found (yet) in clinical settings mostly from ethical reasons. The potential mechanisms of COVID vaccine-related anaphylaxis or cytokine storm are better studied and broader then they claim. The role of PEG and complement activation is among them just to mention two of them. Therefore, I believe the authors should present a more complete overview. Without being exhaustive, I mention some relevant studies just for orientation.
Role of anti-polyethylene glycol (PEG) antibodies in the allergic reactions to PEG-containing Covid-19 vaccines: Evidence for immunogenicity of PEG, VACCINE 41: (31) pp. 4561-4570.
A naturally hypersensitive porcine model may help understand the mechanism of COVID-19 mRNA vaccine-induced rare (pseudo) allergic reactions: complement activation as a possible contributing factor, GEROSCIENCE: 44: (2) pp. 597-618.
Allergic Reactions and Anaphylaxis to LNP-Based COVID-19 Vaccines, Mol Ther. 2021 Mar 3;29 (3):898-900. doi: 10.1016/j.ymthe.2021.01.030. Epub 2021 Feb 5.
Comments on the Quality of English LanguageEnglish Language requires minor edition only.
Reviewer 3 Report
Comments and Suggestions for Authors
The manuscript presents a review describing mechanistic aspects of two major acute adverse effects following a vaccination applying COVID-19 mRNA-LNP vaccines as well as presenting a suitable approach for a correct diagnosis and a promising treatment. As major conclusions, the monitoring of IL-6 levels and the administration of corticosteroids to distinguish between cytokine storms and anaphylaxis post-vaccination are elaborated. This review offers a concise, timely description for the diagnosis and treatment of acute adverse effects as a consequence of COVID-19 mRNA-LNP vaccines. Since the detrimental effects of the world-wide vaccination campaign employing the above mentioned vaccines, resulting in an unprecedented number of vaccinees with adverse effects, who were physically harmed or died, are meanwhile unambiguously documented, this review is of interest to the readership of Diseases.
Before acceptance, some amendments are recommended:
1. The first sentence in the Introduction needs to be specified.
1a. Are other deaths at a later stage after vaccination with COVID-19 mRNA-LNP vaccines considered?
1b. In many studies, particulary those which were an essential support to obtain and secure the EUAs from the authorities, omitted adverse effects including deaths within a certain time window after vaccination shots, e.g. two weeks. Those vaccinees were frequently counted as unvaccinated. The notorious underreporting of acute adverse effects including death was earliest proven in November 2021 by the court-enforced release of hitherto undisclosed Pfizer studies. For example, refer to the Pfizer study summarised in a document (https://phmpt.org/wp-content/uploads/2021/11/5.3.6-postmarketing-experience.pdf) titled "CUMULATIVE ANALYSIS OF POST-AUTHORIZATION ADVERSE EVENT REPORTS OF PF-07302048 (BNT162B2) RECEIVED THROUGH 28-FEB-2021" stating that vaccine effectiveness is not proven as well as listing thousands of patients' cases with adverse effects (including death).
In section 2.1, the faulty design of COVID-19 mRNA-LNP vaccines needs to be discussed. In particular, the damaging effects of the ionizable cationic lipid ALC-0315 used in Pfizer/BioNTech's Comirnaty vaccine as a consequence of an unsuitable pKa are documented (Refer to Segalla (2023) Apparent Cytotoxicity and Intrinsic Cytotoxicity of Lipid Nanomaterials Contained in a COVID-19 mRNA Vaccine. International Journal of Vaccine Theory, Practice, and Research , 3(1), 957-972. https://doi.org/10.56098/ijvtpr.v3i1.84). Furthermore, the detrimental effects of modified mRNA bases used in the COVID-19 mRNA-LNP vaccines, such as their cancerogenic potential are long known (e.g. Rubio-Casillas et al. (2024) Review: N1-methyl-pseudouridine (m1Ψ): Friend or foe of cancer? Int J Biol Macromol. 270(Pt 2):132447. doi: 10.1016/j.ijbiomac.2024.132447.).
2. Intravenous application of high ascorbic acid doses is a safe and effective treatment to cure acute disease conditions, such as sepsis, as ascorbic acid represents a prime antitoxin, generated by many organisms. This is well documented in thousands of publications, which remain essentially ignored so far in the health sector. This should be briefly discussed at the end of section 3.
3. The first sentence in the Discussion should rather read ".... since 2015".
Comments on the Quality of English Language
Refer to report.
Round 2
Reviewer 2 Report
Comments and Suggestions for Authors
N/A
Comments on the Quality of English LanguageN/A